# Type 2 Diabetes Prevention Programs—From Proof-of-Concept Trials to National Intervention and Beyond

**DOI:** 10.3390/jcm12051876

**Published:** 2023-02-27

**Authors:** Jaakko Tuomilehto, Matti Uusitupa, Edward W. Gregg, Jaana Lindström

**Affiliations:** 1Population Health Unit, Finnish Institute for Health and Welfare, Mannerheimintie 166, 00271 Helsinki, Finland; 2Department of Public Health, University of Helsinki, 00100 Helsinki, Finland; 3Diabetes Research Group, King Abdulaziz University, Jeddah 21589, Saudi Arabia; 4Institute of Public Health and Clinical Nutrition, University of Eastern Finland, 70210 Kuopio, Finland; 5School of Population Health, Royal College of Surgeons of Ireland, University of Medicine and Health Sciences, D02 YN77 Dublin, Ireland; 6Department of Epidemiology and Biostatistics, School of Public Health, Imperial College, London SW7 2AZ, UK

**Keywords:** diabetes prevention, proof-of-concept trials, national implementation, impaired glucose tolerance

## Abstract

The prevention of type 2 diabetes (T2D) in high-risk people with lifestyle interventions has been demonstrated by several randomized controlled trials. The intervention effect has sustained up to 20 years in post-trial monitoring of T2D incidence. In 2000, Finland launched the national T2D prevention plan. For screening for high T2D risk, the non-laboratory Finnish Diabetes Risk Score was developed and widely used, also in other countries. The incidence of drug-treated T2D has decreased steadily since 2010. The US congress authorized public funding for a national diabetes prevention program (NDPP) in 2010. It was built around a 16-visit program that relies on referral from primary care and self-referral of persons with either prediabetes or by a diabetes risk test. The program uses a train-the-trainer program. In 2015 the program started the inclusion of online programs. There has been limited implementation of nationwide T2D prevention programs in other countries. Despite the convincing results from RCTs in China and India, no translation to the national level was introduced there. T2D prevention efforts in low-and middle-income countries are still limited, but results have been promising. Barriers to efficient interventions are greater in these countries than in high-income countries, where many barriers also exist. Health disparities by socioeconomic status exist for T2D and its risk factors and form a challenge for preventive interventions. It seems that a stronger commitment to T2D prevention is needed, such as the successful WHO Framework Convention on Tobacco Control, which legally binds the countries to act.

## 1. Introduction

The need for the prevention of type 2 diabetes (T2D) was first discussed in published scientific discourse 100 years ago [1], but rigorous randomized controlled trials (RCTs) to prevent T2D were not conducted before the 1990s. Earlier prevention attempts were small and did not apply experimental study designs [2], leaving us to rely on ecological and observational study evidence for decades to guide knowledge about the role of lifestyle factors in the development of diabetes. The slow progress with population studies and RCTs was due to several factors, including the lack of agreed-upon definitions [3,4], the limited application of behavior change interventions in clinical settings where trials are historically conducted, and evidence-based medicine only developed momentum in the 1970s. Thus, T2D had already reached epidemic proportions in some populations before the discussion about the need for the primary prevention of T2D was seriously initiated [5].

Successful prevention of any disease requires several essential prerequisites: agreed diagnostic criteria, knowledge about risk factors and natural history of the disease, affordable and acceptable screening methods to identify high-risk individuals, and acceptable and efficient methods to influence modifiable risk factors. The next crucial step is proof-of-concept trials to demonstrate the effects of interventions on risk reduction in high-risk people. If interventions are proven effective, they can be implemented at the population level, and national prevention policies can be developed. The proof-of-concept RCTs have unequivocally shown that the onset of T2D in high-risk people can be successfully postponed [6,7,8,9,10].

Several policy-level recommendations helped pave the way for stronger T2D prevention science. The 42nd World Health Assembly in 1989 [11] adopted the resolution on prevention and control of diabetes stating: “… invites Member States: (i) to assess the national importance of diabetes, and (ii) to implement population-based measures, appropriate to the local situation, to prevent and control diabetes”. In 1994, the WHO Study Group report entitled “Prevention of diabetes mellitus” was published [12]. However, these documents were published prior to the proper RCT evidence on the efficacy of the prevention of T2D. In 2011, the United Nations Summit on Non-Communicable Diseases outlined the main targets for 2025; the change targets for obesity and diabetes were conservative at 0%, while for hypertension, the target was a 25% reduction [13].

In this narrative review, we summarize the original findings from the main proof-of-concept RCTs and discuss how their findings have been translated to real-life settings. In addition, we will discuss issues regarding barriers and opportunities in scaling up interventions to the national or regional level and what the main requirements are for a successful T2D prevention program.

## 2. First RCTs Considered as the Proof-of-Concept Trials on T2D Prevention

The main proof-of-concept RCTs [6,7,8,9,10] convincingly documented that the progression from IGT to T2D can be prevented by lifestyle intervention (Table 1) and summarized in review articles [11,14]. This has been a breakthrough in diabetes research.

The Da Qing Diabetes Prevention Study carried out in northern China had 33 participating clinics that were randomized to carry out the intervention according to one of the four intervention protocols (diet alone, exercise alone, diet-exercise combined, or none); trial participants were not individually randomized [6].

The Finnish Diabetes Prevention Study (DPS) was the first proper RCT on the prevention of T2D with lifestyle modification with individual randomization. Notably, none of the participants who adhered fully to the lifestyle intervention and reached four or five of the five lifestyle targets developed T2D during the trial [7]. The Diabetes Prevention Program in the USA (DPP) compared the efficacy and safety of three interventions: an intensive lifestyle intervention or standard lifestyle recommendations, combined with metformin or placebo using individual randomization [8]. During the RCT phase, a 58% reduction in T2D risk was seen by lifestyle intervention both in the DPS and DPP.

In the Japanese Trial of Men with IGT, participants were randomly assigned in a 4:1 ratio to a standard intervention group or intensive lifestyle intervention group [9]. The Indian Diabetes Prevention Programme (IDPP) randomized middle-aged people with IGT into four groups: (1) the control group, (2) lifestyle modification advice (LSM), (3) metformin (MET), and (4) LSM plus MET [10].

These RCTs have provided a large amount of new knowledge and understanding regarding the potential for the prevention of T2D in high-risk people with IGT, as summarized in Table 2.

Since all these trials included people with IGT, the efficacy in other people at an elevated risk of T2D remained unknown. In obese people, weight reduction along with increased physical activity and healthy dietary choices, were applied as multimodal prevention tools [15]. Dietary approaches have been relatively similar, emphasizing energy restriction, reduction of intakes of total fat and saturated fatty acids, and increasing vegetable, fruit, and whole grain intakes. One study was based on the Mediterranean diet rich in virgin olive oil or nuts [16]. Data from lifestyle interventions on RCTs based on other prediabetic phenotypes such as IFG or elevated HbA1c have been inconsistent [17,18]. The Japanese RCT showed that the possible favorable effect on T2D incidence in people with IFG was restricted to those who also had IGT at baseline [17]. Recently, it has been shown that prediabetes is phenotypically heterogeneous not only by glycemic markers but other factors as well, and the progression to T2D may vary due to non-glycemic markers [19].

## 3. Long-Term Effects of Preventive Interventions in the T2D Prevention RCTs

Some RCTs have carried out a follow-up of the trial participants after the trial. The meta-analysis of the incidence of T2D during the post-trial non-intervention period showed a 20% risk reduction [15,20]. This estimate was strongly influenced by the 6% only risk reduction during the DPP Outcomes Study (DPPOS), where all participants, regardless of their original treatment group, were offered lifestyle counseling [21].

The Da Qing trial participants were followed for 30 years from the trial onset [22,23]. The sustained 49% benefit on the T2D incidence was seen during the non-intervention follow-up period in people who had received lifestyle intervention. In the DPS, people free of T2D at the end of the active intervention period continued to attend the annual clinical examinations, but the intervention ceased. The effect of the lifestyle counseling was sustained, and an additional 36% risk reduction during the post-intervention period after 3 years and a 39% reduction after nine years was observed [24,25]. Questions remain about the degree to which lifestyle interventions for diabetes prevention also convert into reduced risk of diabetes complications. None of the proof-of-concept trials were statistically powered to study cardiovascular (CVD) outcomes. Thus, any post hoc findings must be reviewed with caution. The Da Qing diabetes prevention follow-up study found reduced rates of CVD events and deaths and composite microvascular events 24 years after the completion of the intervention among persons in the interventions [22]. In the DPS, the occurrence of early retinopathy changes, i.e., microaneurysms, was significantly lower in the intervention than in the control group [26]. In DPS, total mortality and CVD incidence did not differ significantly between the intervention and control groups during the extended 10-year follow-up [27]. However, compared with the population-based cohort with IGT, adjusted HRs were 0.21 (95% CI 0.09–0.52) and 0.39 (0.20–0.79) for total mortality, and 0.89 (0.62–1.27) and 0.87 (0.60–1.27) for cardiovascular morbidity in the intervention and control groups, respectively. The risk of death in the DPS combined cohort was markedly lower than in the Finnish IGT cohort (adjusted HR 0.30, 95% CI 0.17–0.54), while the difference in the risk of CVD was smaller (adjusted HR 0.88, 95% CI 0.64–1.21). A large proportion of the DPS participants received evidence-based management for hypertension and dyslipidemia both in the intervention and control group since they were advised to contact their physician if values were too high at the annual visit. In DPPOS, neither metformin nor lifestyle intervention reduced major cardiovascular events over 21 years despite long-term prevention of diabetes [28,29]. Provision of group lifestyle intervention to all, extensive out-of-study use of statin and antihypertensive agents, and reduction in the use of study metformin together with out-of-study metformin use over time have diluted the effects of the DPP interventions. Additionally, since the modern management of T2D has emphasized a multifactorial treatment of people with diabetes, in the control groups of the DPS and DPP have received treatment for hypertension and dyslipidemia once they had been diagnosed with T2D, thus earlier than in the intervention group where people were free of T2D longer time.

Although the issue of preventive effect on complications remains under debate, future T2D prevention studies should also consider the possibility that some of the benefits of lifestyle intervention will lie in more diverse outcomes, such as disability, sleep apnea, quality of life, depression, and mental health [30].

## 4. Lifestyle Outcomes

We present here how the lifestyle counseling in the DPS was planned; it aimed at a healthy diet and physical activity as generally recommended and weight reduction >5% from baseline weight since all participants were overweight/obese. The participants in the intervention group improved their diet and increased moderate-to-vigorous physical activity more than control group participants, and they lost more weight, 5.1% vs. 1.2%, respectively, after the first year [31]. Achievement of the lifestyle goals was directly associated with T2D risk reduction during the intervention period [7]. Importantly, four years after the discontinuation of the counseling, the former intervention group participants still consumed a diet with lower fat and saturated fat and higher fiber content than the control group participants. Both the intervention and control group participants experienced a gradual weight regain during the follow-up, but the former intervention group participants were still slightly below their baseline weight after 10 years [25]. This proves that sustained lifestyle change is possible to achieve because of professional, long-lasting support. To our knowledge, the other major diabetes prevention trials have not reported the participants’ diet or physical activity during the post-intervention follow-up period.

## 5. Translation of the Findings from the RCTs to Real-Life Settings

### 5.1. Implementation Strategies in Finland

In Finland, a nationwide program, The Development Programme for the Prevention and Care of Diabetes (DEHKO 2000–2010), was established in 2000. Its main aim was to improve the quality of diabetes care in Finland, increase awareness of the worsening diabetes epidemic, and enhance the activities of both public stakeholders and Finnish citizens in the prevention and treatment of diabetes. Although one of the goals of the DEHKO was to establish a primary prevention program for T2D in the entire country, an invitation to participate in the prevention program was sent to all 20 Finnish hospital districts, of which five with a population of 1.6 million people (29% of the Finnish population) were willing to participate in the special implementation program (The Finnish National Prevention Programme, FIN-D2D). This represented the first large-scale T2D prevention implementation program in the world and was carried out within primary healthcare settings between 2003 to 2008 [32]. There was a common guideline for prevention activities, but each hospital district developed its individual implementation plan. The main funding for the FIN-D2D program came from the Ministry of Social Affairs and Health, which contributed 8.4 million Euros. Additional funding came from local hospital districts, municipalities, and the Finnish Diabetes Association, along with private companies that financially supported the program.

One of the main successes of the FIN-D2D was the development of the non-laboratory Finnish Diabetes Risk Score (FINDRISC) for an easy screening for potential high-risk individuals [33]. The FINDRISC was used in primary healthcare and occupational healthcare clinics nationwide. The FINDRISC has also been validated and used in many countries worldwide. In the Finnish national T2D prevention action, principles of interventions were based on the DPS experiences, but the actual details of the implementation were left to the local stakeholders. In line with the data from the RTCs, the prevention effect was tightly correlated with body weight reduction; in the people with ≥5% body weight loss (17.5%) at the 1-year examination, the risk of diabetes was 69% lower than in those with stable body weight, and even modest of 2.5–4.9% weight reduction lowered risk of diabetes by 28% [34]. The long-term results of the FIN-D2D with an average follow-up time of 7.4 years showed that the risk reduction of drug-treated diabetes remained significantly reduced (approximately 30%) in individuals with a weight reduction >2.5% after one year [35].

We have monitored the incidence of drug-treated diabetes in Finland. The age-standardized annual drug reimbursement rates for new glucose-lowering drugs in people aged 40–79 years in the five FIN-D2D areas and in the rest of Finland increased from 2000 to 2010 and then decreased from 2010 to 2020 (Figure 1). This increase might be partly due to more active screening of diabetes promoted in the DEHKO 2000–2010 and FIN-D2D. The specific impact of the FIN-D2D program per se after 2010 is difficult to estimate due to the huge increase in awareness of diabetes in the Finnish population in general.

### 5.2. Development of National Programs in the US

In 2010, the US congress authorized public funding for a national diabetes prevention program to be coordinated by the US Centers for Disease Control and Prevention. The impetus for the support and for the strategies that followed stemmed from three general sources of evidence [36]: First, as summarized above, the major trials of diabetes prevention showed a strong reduction in incidence over 2–6 years provided the proof-of-concept for the benefits of a multi-disciplinary behavioral approach (6–10). A subsequent systematic review showed a pooled 28% reduced incidence across 5 to 10 years after the completion of the intervention, indicating that the prevention effect was highly sustainable (20–25). A systematic review and network meta-analysis of the real-world impact on incidence, weight, and glucose suggested an absolute benefit in the reduction of diabetes in controlled studies during the first 1–2 years was approximately 3% with a significant relative risk reduction of 0.71 (95% CI 0.58, 0.88) [37]. In analyses combining controlled and uncontrolled studies, participants receiving group education by healthcare professionals had a 33% reduced diabetes risk (odds ratio 0.67 [0.49, 0.92]). Intervention participants lost 1.5 kg more weight and achieved a 0.09 mmol/L greater FPG decrease than control participants. Every additional kilogram lost by participants was associated with 43% lower diabetes odds (b = 0.57 [0.41, 0.78]). In the proof-of-concept trials, the absolute risk reduction has been approximately 17%. Reasons for the lower absolute benefits in real-world studies compared with the proof-of-concept trials are several: differences in entry criteria, intensity and design of interventions, duration of interventions, etc.

Second, the evidence from community translation trials during the 2000s and into the 2010s established the feasibility of the delivery of effective approaches outside of idealized research settings. These studies have shown that an average of 3–4% weight loss, equivalent to about two-thirds of the weight loss efficacy seen in controlled trials, can be consistently achieved in interventions conducted in diverse community settings [38]. Community programs have also observed significant improvements in blood pressure and fasting blood glucose [39]. Studies of risk factor changes in studies have established favorable cost-effectiveness, particularly if higher-risk participants can be efficiently identified [40,41]. Additionally, costs have been lower when programs have been delivered to groups in community or primary care settings [40,41].

The US National DPP (NDPP) was established around the scientific principles established by the prevention trials, including the importance of multi-disciplinary behavioral support, aiming for goals of moderate weight loss, a healthy diet, and increasing physical activity levels [42,43,44]. The US NDPP was initially built around a 16-visit program over 6 months, 6 additional maintenance sessions, and a weight loss goal of 5–7%. The program relies on both referrals from primary care and self-referral to community programs for persons with either prediabetes based on glycemic tests (ADA definition), or by a diabetes risk test. The program uses a train-the-trainer program wherein “master trainers” provide training and support to “trained lifestyle coaches”, who then provide prevention services based on a “recognized program”. Program recognition is achieved through the application and continued reporting of basic data to the CDC’s “Diabetes prevention recognition program”, which also maintains a central data source of programs and participant response parameters. The program effort involves streams of work to build a maintain a workforce of master trainers and lifestyle coaches to deliver programs, continue updating the curriculum, promote the continued development of programs, and promote referrals and engagement. In 2015, the program started the inclusion of online programs, and in 2018, the Center for Medicare and Medicaid Services (CMS) became an additional provider of NDPP programs.

The first published findings from the US NDPP showed a median weight loss of 5% among participants, with a clear dose response relationship between the number of sessions attended and the magnitude of weight loss [45]. By February 2022, the US NDPP consisted of 2126 programs and more than 595,654 persons enrolled (E. Gregg, personal communication), with the primary growth in the program participation occurring through the online service.

Several major challenges remain for these national programs. In the US, the US Preventive Services Task Force recommends that adults aged 35–70 who are overweight or obese be screened for pre-diabetes and that clinicians offer or refer patients with pre-diabetes to effective programs [46]. The CDC-recognized programs require a blood test, and ADA-defined pre-diabetes, for at least one-third of participants in programs and risk score for the remaining two-thirds. The Medicare NDPP requires a blood test and slightly more specific thresholds (FPG > 110 mg/dl; 2-h glucose > 140 mg/dl, or HbA1c > 5.7%). With these broad recommendations for both screening and eligibility in place, the limiting factors to participation appear to rest more on availability, reimbursement, engagement, and participation. At present, reimbursement is variable across the country, existing for persons with CMS, but has variable coverage for the important segment of the population less than 65 years of age. Further, disadvantaged populations and young adults have particularly low rates of testing and engagement [47]. These challenges underscore the importance of continued examination of real-world cost-effectiveness analyses and continued refinement of programs that can tailor interventions to different high-risk groups to ensure effectiveness and optimize long-term adherence and maintenance of a healthy lifestyle.

### 5.3. Dissemination of Experiences from the Proof-of-Concept Trials to Other European Countries

The European Diabetes Prevention Study (EDIPS) collaboration applied the DPS protocol in other European countries, e.g., the SLIM study in Maastricht, The Netherlands, and the EDIPS Newcastle study in Newcastle upon Tyne, UK, in addition to the DPS [48]. The pooled results showed a 57% reduction in T2D incidence. Thus, the DPS approach was externally validated successfully.

The DE-PLAN (“Diabetes in Europe–Prevention using Lifestyle, Physical Activity, and Nutritional intervention”) project was established after the success of the DPS and setting up the FIN-D2D project in Finland to address the major public health concern of T2D in Europe [49]. The DE-PLAN project aimed at developing and testing models of efficient identification of individuals at high risk of T2D diabetes in the community using the FINDRISC. In addition, lifestyle intervention practices in people at high risk of T2D were developed within existing healthcare systems in 17 European countries. The FINDRISC is probably the most validated and used diabetes risk score, either in the original form or with some modifications. In PubMed, 214 articles were identified searching for FINDRISC in April 2022.

It should be kept in mind that the ultimate aim of T2D prevention is to improve health in people at high risk of T2D. The DE-PLAN study demonstrated that a community-based T2D prevention program based on lifestyle intervention might lead to substantial improvements in health-related quality of life [50]. Since the intervention was implemented on diverse populations and through different strategies, these findings may serve as a realistic perspective of tackling the diabetes epidemic across Europe.

A comprehensive European guideline [51] and a toolkit [52] for the prevention of T2D (IMAGE) were prepared in 2010 with European Union support. These have been used in many countries as the basis for various T2D prevention activities, and methods for systematical follow-up, evaluation, and quality indicators for diabetes prevention were developed [53].

Beginning in 2016 (about 4 years after the US), the UK National Health Service initiated a similar program based on a 9-month, 13-session curriculum designed for 16 h of contact [54] in people with non-diabetic hyperglycemia or gestational diabetes within the prior 12 months. As of 2022, more than 800,000 adults had been referred. The UK program found an average weight loss of 3–5 kg and an HbA1c reduction of 1.3 mmol/mol after one year. A recent review included 65 articles reporting on the English NHS DPP since 2015 that were eligible for inclusion, most of them published between 2018 to 2020 [55]. The articles reported on uptake and retention, implementation considerations, program outcomes, stakeholder experience, and screening and referral processes, various research methods. Articles revealed preliminary evidence on service user characteristics, rates of referral, uptake, and retention, as well as how far the NHS DPP is being delivered in line with its evidence base and service specification. The evidence is gradually accumulating on NHS DPP uptake and retention, with emerging evidence on program outcomes such as weight loss and HbA1c.

There is continued debate around the optimal risk stratification policies for diabetes prevention. Systematic reviews of short-term studies have shown that lifestyle interventions can improve different glycemic markers across the full spectrum of risk [56]. Although the proof-of-concept trials that tested the impact on T2D incidence focused on people with IGT [6,7,8,9,10], subsequent RCTs and community translation trials used more liberal inclusion criteria, including persons with isolated IFG, elevated HbA1c, and non-biochemical risk scores [38,39,40]. However, the degree to which this liberalization of criteria results in reduced effectiveness of interventions remains unclear. A post hoc analysis of the Japanese DPP found that significant preventive effects were driven by the subgroup with IGT [17]. In the Diabetes Community Lifestyle Improvement Program (CLIP), there was evidence for heterogeneity of the intervention effect across prediabetes types, with the strongest benefit in people with combined IFG + IGT (36%) and isolated IGT (31%) and not significant in isolated IFG (12%). [57] The randomized controlled trial, the Kerala Diabetes Prevention Program, a low-cost community-based peer-support lifestyle intervention where participants were identified on the basis of a risk score, and the majority had isolated IFG or NGT, resulted in a nonsignificant reduction in diabetes incidence [58]. In general, including persons of lower risk will increase the overall number of cases prevented but reduce efficiency and cost-effectiveness by spending resources on persons with a relatively low risk of progression to diabetes [59].

### 5.4. Prevention of T2D in Low- and Middle-Income Countries

Most but not all trials and major implementation programs of T2D prevention have been limited to high-income countries. Despite encouraging progress in the several countries noted above, the proportion of high-risk adults reached by these programs remains low, major challenges remain in referral, uptake, and engagement, and the long-term health outcomes of these programs have not yet been evaluated. This presents a major gap in both science and implementation because 80% of the world’s cases of T2D now reside in low- or middle-income countries [60]. An increasing number of studies are being conducted in LMICs, generally showing the benefits of lifestyle interventions. However, debate remains around the degree to which implementation programs should employ individual-based approaches using community health worker models as opposed to focusing solely on population-wide approaches [61,62]. In high-income countries, resources exist for the high-risk approach, whereas in LMICs, healthcare resources are limited for preventive measures on an individual basis. These questions underscore the need for well-designed large-scale implementation studies and natural experiments of ongoing policies (such as the Mexico soft drink tax example).

Non-governmental organizations (NGOs) can have various health sector functions such as service provision, social welfare activities, support activities, and research and advocacy. With these actions, they make an important contribution to healthcare, in particular to disease prevention, either alone or jointly with the local healthcare systems [63].

A systematic review and meta-analysis summarized and quantified the evidence published over the last two decades regarding the effectiveness of lifestyle interventions for the prevention of T2D and changes in cardiometabolic factors among 48 at-risk populations in LMICs [64]. Most of these studies were published in the last 10 years and contribute to reducing the research gap regarding the availability of evidence coming from settings with constrained resources. Different intervention methods have been applied in these studies. Comprehensive, multi-target, and multi-component lifestyle interventions with a median duration of 18 months reduced the T2D incidence in 14 studies by 25% on average. Improvements in glycemic levels, weight, and cardiometabolic indicators were also found to be associated with lifestyle interventions. Thus, this review confirmed previous observations from RCTs in high-income countries on the efficacy of lifestyle interventions used as preventive strategies for T2D. It has been previously reported that culturally tailored and targeted interventions yield better results than a generalized approach to preventing T2D [65,66].

The reach of lifestyle interventions in LMICs will be greatly limited if they depend on scarce and expensive professional healthcare workers and if participant time and travel costs are barriers to access. With respect to intervention feasibility, results indicated that community health workers were able to deliver the sessions and that participants attended at acceptable levels in studies in Kerala, India [58], Grenada [67], and South Africa [68].The video-based design used in South Africa avoids the need for expert involvement and is thus highly scalable and could also be delivered via smartphone or an online platform. Such interventions may be recommended for widespread scale-up, but they need to be adapted to the local settings, and their feasibility and generalizability must be tested in various countries and settings.

A comprehensive review of barriers and facilitators related to the management of diabetes, hypertension, and dyslipidemia, considering patient, health provider, and health system perspectives [69] that exist in high-income countries but especially in LMCIs. These include administrative requirements, i.e., confusion about how the system works and bureaucratic processes for accessing health programs. The lack of support and communication from professionals towards patients, very short consultations, inadequate or lack of counseling, clinical inertia, lack of communication between health staff members (ambiguity regarding responsibilities, no team approach to care), negative attitude or mistreatment by staff, lack of continuity with the same healthcare professional, and infrequent follow-up/monitoring. In addition, inadequate staff incentives and training, inadequate logistic systems, inadequate infrastructure, long wait times, lack of coordination between different providers (e.g., public and private), lack of trust and dissatisfaction with the treatment provided by healthcare providers, inadequate information technologies, etc. are common problems in LMCIs. Additionally, a lack of knowledge of the disease, illiteracy, and low education can contribute to a poor understanding of the disease and the need for action. Due to the fact that T2D and prediabetes are asymptomatic conditions, people with such disorders may consider interventions unnecessary. Geographic accessibility with long distances to healthcare facilities both in cities and rural areas and the cost of transportation form important barriers.

### 5.5. T2D Prevention in Youth

The prevalence of T2D has been reported to increase among youths, but there are very few studies on T2D prevention among youth at high risk [70]. The Yale Bright Bodies study, an RCT, tested an adapted DPP among youths with prediabetes [71]. Reductions in 2-h glucose following a 6-month intervention were larger than in the standard clinical care group. The emphasis was on weight management maintenance; the difference between groups was 3 kg (*p* = 0.006), and adiposity (BMI z score (*p* < 0.001) was reduced compared with the control group. Another small study tested a culturally designed T2D prevention intervention in US Latino youths with prediabetes. Both intervention and usual care reduced T2D risk factors to a similar degree [72].

The primary/primordial prevention of T2D and prediabetes should start at an early age and at a time when metabolic abnormalities have not yet developed. The first study on the long-term effects of combined physical activity and dietary intervention on insulin resistance and fasting plasma glucose was carried out in a general population of predominantly normal-weight Finnish children aged 6–9 years at baseline [73]. The effects were assessed on insulin, glucose, and HOMA-IR using the intention-to-treat analysis after adjustment for sex, age at baseline, and pubertal status at baseline and 2-year follow-up. The combined physical activity and dietary intervention attenuated the increase in insulin resistance. This beneficial effect was partly mediated by changes in physical activity, sedentary time, and diet but not changes in body composition. These results are promising for early intervention to reduce the T2D risk in children. However, it is challenging to find out how such interventions could be implemented nationwide or globally.

### 5.6. Challenges to Implement T2D Prevention in High-Risk People in Real-Life

A systematic review evaluating the implementation of real-world diabetes prevention programs between 2001 to 2015 was published, which suggested that while a high-frequency intervention plays an important role in achieving high weight loss outcomes, some programs with lower-intensity interventions also showed reductions in the incidence of T2D [74]. Given the increasing prevalence of T2D, and inequitable access to healthcare resources, there is a need for approaches to develop diabetes prevention programs that can provide care to assist those at the highest T2D risk. The rapid adoption of technology and wide acceptability of remotely delivered health-related interventions has shifted T2D prevention towards electronic and mobile health (eHealth and mHealth, respectively) solutions to augment and improve access to care and save costs for the participants. Such programs applied in T2D prevention were recently summarized by McPherson et al. [75]. Although these approaches seem to have been well-received by participants, the influence of mHealth interventions has had mixed effects on behavioral outcomes and T2D incidence. Nevertheless, it is likely that mHealth interventions have promising potential, especially in LMICs. They may be used both for individual and group interventions. Many applications developed by both commercial and healthcare providers already exist, but they must be tailored to the local cultural, religious, and societal settings. Additionally, if they are used as a part of regional or national T2D prevention plans, they need to be approved by the local healthcare system and implemented in a coordinated fashion. mHealth interventions may fit well for the NGO’s working concept. Several real-world studies on T2D prevention have demonstrated an important challenge with the high-risk approach in non-communicable disease (NCD) prevention: the identification of the high-risk status does not lead to the adoption of services available for risk reduction. For instance, in the English NHS DPP, 324,699 people were referred to the program, of whom 152,294 attended an initial assessment, and only 96,442 (30%) attended at least one of the group-based intervention sessions [54]. In the Finnish D2D study, 10,149 individuals at high risk for T2D were identified as primary healthcare centers, but one-year follow-up data were available only for 2798 (28%) participants [34].

### 5.7. Precision Medicine in the Prevention of T2D

Currently, over 500 hundred genetic variants have been identified to be associated with the risk of T2D [76]. Most of them are involved in insulin secretion, and their individual impact is small. Beyond genetic and phenotypic differences, the risk of T2D and prediabetes varies according to epigenetic factors and metabolite profiles [77,78]. Wareham pointed out that precision medicine approaches are promising, but they need longer-term evaluation against clinical outcomes. Whatever personalized preventive approaches for T2D are developed in the future, they will need to be complementary to existing individual-level interventions that are being rolled out and that are demonstrably effective [79].

Gene-diet interaction analyses are available from the DPS and DPP studies. In the DPS, lifestyle intervention was equally effective in individuals with a high genetic risk as compared with those with lower genetic risks [80]. Several post hoc analyses based on the data of DPP on the interactions between lifestyles and genetic variants on the incidence of T2D have been published, but no consistent interactions have been reported [81]. The recently initiated Finnish T2D-GENE trial may provide additional knowledge to this question [82].

In epidemiologic studies, a low dietary fiber intake is constantly associated with a higher risk of diabetes. Since, in many westernized countries, whole grain and fiber intakes remain low, there are good reasons to suggest that low-fiber diets may contribute to the current epidemic of diabetes [83]. Dietary fiber may modify glucose absorption from the gut and contribute to the production of a multitude of metabolites originating from gut microbiota, e.g., bile acids, short-chain fatty acids, several lipid metabolites, and branched-chain amino acids that are known to affect glucose metabolism [84]. In the DPS, increased fiber intake contributed to a lower risk of T2D [31], and a metabolite of tryptophan, indolepropionic acid, was directly associated with fiber intake, a lower risk of diabetes, and better insulin secretion [85]. Thus, increasing the intake of dietary fiber may significantly decrease the risk of T2D, especially in low-income populations, since many high-fiber foods can be inexpensive and have been the basis of local diets during the past. Unfortunately, the current food and soft drink industries are working in the opposite direction, making healthy choices for people difficult. Therefore, societal actions are needed to reduce the T2D epidemic.

### 5.8. Can High-Risk Strategy Be Implemented without a Simultaneous Population Strategy, and What Is Their Cost-Effectiveness?

The WHO and others have recommended taxation as part of a comprehensive package of policy interventions to prevent diet-related NCDs (https://www.euro.who.int/en/health-topics/disease-prevention/nutrition/news/news/2022/3/taxes-on-sweetened-drinks-who-explains-how-to-make-them-an-effective-health-measure; URL accessed on 3 March 2022). Sugar-sweetened beverage (SSB) consumption is one of the modifiable risk factors for NCDs. There is a consistent association between a high intake of SSB and an increased risk of NCDs, including obesity and diabetes. A policy analysis of SSB taxes was recently conducted in the WHO European Region, drawing on theories of policymaking and diffusion of innovation [86]. Data were collected from policy documents and media, secondary contextual sources, and qualitative interview data (n = 20) to analyze factors influencing the adoption of taxes in 10 countries Belgium, Finland, France, Hungary, Ireland, Latvia, Monaco, Norway, Portugal, and the UK that had current SSB taxes. Norway and Belgium adopted SSB taxation in 2009, followed by Finland from 2011 to 2017. It was noted as a potential NCD health strategy in Finland, Ireland, and Portugal. In the US, some states and cities have also adopted SSB taxes. It has been proposed that taxation may be a cost-effective strategy for reducing sugar intake, obesity, and health costs and generating revenue that can be used for health programs [87]. An excise tax of one peso per liter on sugar-sweetened beverages was implemented in Mexico in 2014 [88]. A cohort simulation model was used to project the impact of the tax over ten years. The tax was estimated to prevent 239,900 cases of obesity and 61,340 cases of T2D, leading to gains of 55,300 quality-adjusted life-years and averting 5840 disability-adjusted life-years. The tax was estimated to save USD 3.98 for every dollar spent on its implementation.

Population-based prevention focuses on strategies to alter the overall distribution of risk in the population. The individual benefits of policy measures are relatively small, but since the effects of such interventions can move the risk factor distribution of the entire population to a lower level, the overall impact on the disease outcome risk can be larger than with interventions restricted to high-risk individuals alone. A recent review identified 39 T2D prevention studies where the cost-effectiveness had been evaluated: 28 targeting high-risk individuals and 11 targeting whole populations [59]. Both lifestyle and metformin interventions in high-risk individuals were cost-effective from a healthcare system or a societal perspective. Compared with lifestyle interventions delivered one-on-one or by a health professional, those offered in a group setting or provided by a combination of health professionals and lay health workers had lower incremental cost-effectiveness ratios. Among population-based interventions, taxing sugar-sweetened beverages was cost-saving from both the healthcare system and governmental perspectives. Subsidies for fresh fruit and vegetable, providing healthy foods in low-income areas and workplace settings, and promotion of physical activity may all be cost-saving or highly cost-effective (Table 3). Although metformin was efficient in reducing the incidence of T2D in the US DPP and Indian DPP, the drug is not registered for the indication of T2D prevention, and in general glucose-lowering drugs should not be a priority in national interventions to prevent T2D.

The analyses of the DPS indicated that the quality of the diet improved, but daily diet costs did not significantly change [89]. The fiber density was inversely associated with diet costs: increasing fiber led to a decrease in diet costs. In Finland, a health economic model utilizing data from multiple sources estimated population-level health economic consequences of increased whole grain food (WGF) consumption among the adult population [90]. Increased WGF consumption could reduce T2D-related costs between 286 € and 989€ million over the next 10 years using the applied scenario with a 10%-unit increase in a proportion of daily WGF users, doubling the number of WGF servings a day or on a combination of these scenarios. Over the next 20–30 years, a population-wide increase in WGF consumption could lead to benefits ranging between 1323 and 154,094 quality-adjusted life years (QALYs) gained due to decreased T2D-related morbidity and mortality during the next 10–30 years.

### 5.9. Who Are the Main Stakeholders in National Interventions to Prevent T2D, and What Resources Are Needed?

To implement T2D prevention at the population level, results from RCTs are important to generate an understanding of the potential for prevention, the possible magnitude of the effect of interventions, and to identify factors that may modify the intervention efficacy. Since T2D is a multifactorial disease, preventive interventions must also be multifactorial. It is also important to emphasize that lifestyle management must be carried out by high-risk people themselves; they cannot “outsource” it to health workers; health workers can only serve as advisors. National interventions to prevent T2D require many stakeholders. Health in All Policies (HiAP) was a term first used in Europe during the Finnish Presidency of the European Union in 2006. It is an intersectoral strategy to include health considerations in policymaking across different sectors such as transportation, agriculture, land use, housing, public safety, and education. It reaffirms public health’s essential role in addressing policy and structural factors affecting health. Stakeholders would also include the local and multinational food and beverage companies against whose products taxes and advertising restrictions are being suggested. Farmers who may produce health-promoting foods might receive subsidies to grow such foods, and their consumption should also be promoted by retailers. Politicians and non-governmental organizations are also relevant stakeholders. An ecological analysis in California found that commuters with the highest distance and time traveled also had the highest rates of physical inactivity and obesity [91]. Similarly, longer travel distances and time have been associated with higher BMI and waist circumference [92]. The prospective Finnish study showed that commuting physical activity was inversely associated with the incidence of T2D [93]. It is commonly agreed that modern city planning should take into account health issues to make physical activities convenient, easy, and safe [94].

### 5.10. Social Determinants of Health in T2D Prevention

T2D is not only a health problem due to the behavior and genetic susceptibility of individuals. Our living environment, whether physical, social, or psychological, may modify people’s risk of T2D and their abilities to apply the lifestyle changes needed to avoid T2D. People exposed to green spaces, especially in their neighborhood, can reduce the risk of being obese and have more physical activity, and reduce their risk of T2D [94]. An inverse association between socioeconomic status (SES) and the prevalence of T2D and its established risk factors is well-known [95]. In the DPP, intensive lifestyle intervention and metformin have greater efficacy among highly educated individuals compared with those with lower educational attainment [96], but in the DPS, lifestyle intervention was effective regardless of participants’ educational attainment [97]. The importance of social determinants of health (e.g., income, education, housing, and access to green spaces and healthy food) and their contribution to health disparities, especially regarding non-communicable diseases, has been recognized for a long time, and various recommendations to reduce them have been made [98]. However, it is important to note that T2D has also increased in people and populations with higher SES, and preventive measures in high-risk people and population levels are important irrespective of SES.

### 5.11. Global Treaty for T2D Prevention Is Necessary

The development of national actions for the prevention of T2D has been disappointingly slow; although RCT results are impressive, the high costs of T2D are well-documented, and the prevalence of T2D has been continuously increasing globally. The reasons for this delay are manifold and remain unclear in many countries. Guidelines on T2D prevention, especially among people at high risk, exist, but it is known that guidelines alone are not sufficient to lead to action. Leadership on T2D prevention at the global level is not visible; for instance, the WHO has not very actively promoted this, although the UN NCD Summit in 2011 included some aspects of diabetes prevention. At the national level, resources for diabetes have been mainly directed to the management of people with T2D, and only limited resources are given to prevention. In addition, the structure of healthcare personnel resources in most countries is predominantly based on physicians and nurses whose training and skills are not necessarily best for lifestyle management. The need for lifestyle coaches, dietitians, specialists in sleep disturbances, etc. is apparent to handle issues related to T2D prevention.T2D is a global epidemic and one of the costliest diseases with high multimorbidity and mortality. Its prevention requires global action. A good example is the WHO Framework Convention on Tobacco Control (WHO FCTC) which is the first treaty on health issues negotiated under the auspices of the WHO. It was approved by the World Health Assembly in 2003 (https://www.who.int/fctc/text_download/en/#:~:text=The%20WHO%20Framework%20Convention%20on,the%20highest%20standard%20of%20health; URL (accessed on 6 March 2022). The WHO FCTC is an evidence-based treaty that reaffirms the right of all people to the highest standard of health. The WHO FCTC was developed in response to the globalization of the tobacco epidemic. There are 182 countries that are parties of the WHO FCTC, and it covers >90% of the world’s population. The WHO FCTC is legally binding in these countries to carry out actions defined by the WHO FCTC. We are calling for a similar treaty for the prevention of T2D that should be developed urgently. This would be important, especially for the development of global population-based strategies for the prevention of T2D. A convention needs to be organized to develop such a treaty. The implementation of its provisions shall be guided by clearly defined principles that may include, as done for the WHO FCTC:-Information on the health consequences of T2D and measures at the appropriate governmental level to protect all persons from exposure to risk factors for T2D;-Strong political commitment is necessary to develop and support initiatives at the national, regional, and international levels, as well as comprehensive multisectoral measures;-International cooperation, particularly transfer of technology, knowledge, financial assistance, and provision of related expertise, taking into consideration local culture, as well as social, economic, political, and legal factors;-Comprehensive multisectoral measures in accordance with evidence-based public health principles;-Technical and financial assistance addressed in the context of nationally developed strategies for sustainable development;-The participation of civil society is essential in achieving the objective of the treaty.

## Figures and Tables

**Figure 1 jcm-12-01876-f001:**
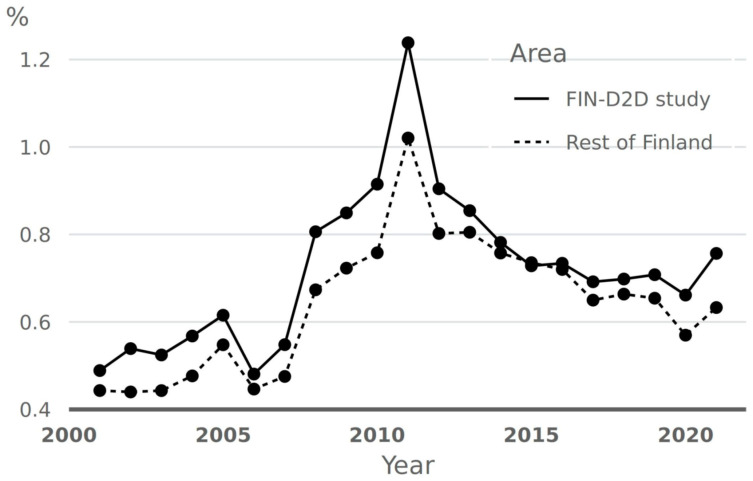
Age-standardized (The Finnish age distribution as standard) incidence of new reimbursement rights for glucose-lowering drug treatment by the Finnish Social Insurance Institution in the FIN-D2D study areas and the rest of Finland.

**Table 1 jcm-12-01876-t001:** Summary of the proof-of-concept trials aimed at preventing the progression to type 2 diabetes in people with impaired glucose tolerance.

	Country; Trial Duration,	Number of Participants	Relative Risk Reduction of Diabetes	Dietary Goals, Weight Change *	Changes in Diet When Available	Physical Activity, Goals/Change	Long-Term Follow-Up
Da Qing IGT and Diabetes Study [6]	China; 6 years	577, all IGT; 33 health clinics	Diet 33%; exercise 47%; diet + exercise 38%	Weight reduction in overweight people; energy restriction; 6-year fall in BMI 1 kg/m^2^ in obese	CHO 58–60 E%; protein 11 E%; fat 25–27 E%; energy decrease 100–240 kcal; BMI goal 23 kg/m^2^	Increase in leisure-time physical activities	Yes
DPS [7]	Finland; 3.2 years	522, all IGT; five centers	58%	Weight reduction >5%; reduce total and SFA; increase dietary fiber; Weight loss 3.5 kg	3-year results: energy reduction 204 kcal; CHO increase 3 E%; fat reduction 5 E%; SFA reduction 3 E%; fiber increase 2 g/1000 kcal	≥4 h/wk; at year 3 sedentary people: 17% in the intervention vs. 29% control group	Yes
DPP [8]	USA; 2.8 years	3234, all IGT + IFG; 27 centers	Lifestyle 58%; Metformin 31%;	Weight loss goal >7%; 1-year weight loss 5.5 kg	Energy reduction 450 vs. 249 kcal and fat intake reduction 6.6 vs. 0.8 E% for intervention and control, respectively.	150 min/wk; 74% reached at 24 months	Yes
IGT trial, Japan [9]	Japan; 4 years	102 in intervention, 356 in control, all IGT	67%	BMI goal 22 kg/m^2^; increase in vegetables; reduce food intake by 10%; fat < 50 g/d; alcohol restriction; Weight loss −1.8 kg	Not reported	30–40 min walking/day	No
IDPP-1 [10]	India; 2.5 years	531, all IGT	Lifestyle 29%; Metformin 26%; lifestyle + Metformin 28%	Reduce total calories, refined CHO, fat and sugar; increase high fiber-rich foods; No change in body weight	Dietary adherence increased in the intervention groups	Walking >30 min a day	No

* weight change difference between the intervention and control groups. IGT = impaired glucose tolerance; IFG = impaired fasting glucose; BMI = body mass index; CHO = carbohydrates; SFA = saturated fatty acids; E% = % of energy intake; DPS: The Finnish Diabetes Prevention Study; DPP: The US Diabetes Prevention Program; IDDP-1: The Indian Diabetes Prevention Programme.

**Table 2 jcm-12-01876-t002:** The summary of the main results from the major proof-of-concept randomized controlled trials (RCTs) to prevent T2D in high-risk people with lifestyle intervention.

1. A very significant and consistent relative risk reduction among the RTCs.
2. The preventive effect of lifestyle intervention was rapid.
3. The benefit was similar in men and women.
4. Lifestyle intervention was effective in all ethnic groups studied.
5. The benefit did not depend on the initial body weight alone, the change in incidence of T2D between the intervention and control groups was parallel regardless baseline BMI that varied among the RCT.s
6. It is not possible to tell which component of the multifactorial intervention contributed most to the preventive effect, but people who managed to reach multiple lifestyle targets benefitted most. Weight reduction is essential in overweight and obese people at high risk of T2D.
7. Lifestyle intervention postponed the onset of T2D by at least for 5 years.
8. A residual risk of T2D existed in the intervention group, primarily due to the lack of success in people reaching lifestyle targets satisfactorily.
9. People with a high genetic risk for T2D benefitted significantly from lifestyle intervention.

**Table 3 jcm-12-01876-t003:** Summary of the cost-effectiveness of whole population policies to prevent T2D (modified from the reference [59]).

Category	Intervention	CE Outcome
**Fiscal policy**		
Sugar sweetened beverage tax	20%, penny-per-ounce, 10%, or $0.5/L tax on Sugar sweetened beverage	Cost saving
Sugar tax	$0.99/100 mL ice cream; $0.9/100 g other products	Cost saving
Subsidy	30% or 0.15/100 g subsidy for fruit/veg consumption	Cost saving to worse health
Combination tax and subsidy	Tax SSB, sat fat., sodium, sugar; subsidy fruit/veg	Cost saving
**Environmental change**		
Fresh food in low-income area	Open supermarket	Cost saving
Workplace healthy food	Provide healthy food in cafeteria	Cost saving
Enhanced physical activity access	Increase facilities for physical activities	$36 k/QALY
**Health promotion**		
Campaign	Community-wide, mass media, or internet campaign to promote physical activity	$87 k/QALY to cost saving
Healthy eating education in low-income community	Diet education and cooking classes	More QALY but no change in cost
Social support PA promotion	Use organized groups to promote physical activity	$35–50 k/QALY
Physical activity promotion for targeted population	Encourage walking and reduce car use using tailored educational information	$17,658/QALY– cost saving

## Data Availability

The study did not require data available statement.

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
