# Peer review of "Type 2 Diabetes Prevention Programs—From Proof-of-Concept Trials to National Intervention and Beyond"

_jcm, 2023, doi:10.3390/jcm12051876_

Round 1
Reviewer 1 Report (Previous Reviewer 1)
You have adequately responded to my points. Though I have no problems with the current version being published as is, I would recommend two minor additions:
1. To add a reference to the comparison of DPPPOS results with a population-based cohort, as I cannot find this in the DPPPOS article that you site.
2. With respect to the NHS DPP, to additionally cite: McManus E, Meacock R, Parkinson B, Sutton M. Population level impact of the NHS Diabetes Prevention Programme on incidence of type 2 diabetes in England: An observational study. Lancet Reg Health Eur. 2022 May 29;19:100420 and comment on its findings.
Author Response
- Line 208 – comment: High-risk strategy interventions, when translated to the community or primary care settings, have demonstrated decreased effectiveness (25-30% in meta-analyses) when compared to the almost 60% risk reduction seen in the original trials.
Authors reply:
We have added text as follows:
A systematic review and network meta-analysis of the real-world impact on incidence, weight, and glucose suggested an absolute benefit in reduction of diabetes in controlled studies during the first 1-2 years was approximately 3% with a significant relative risk reduction of 0.71 (95% CI 0.58, 0.88) (Galaviz). In analyses combining controlled and uncontrolled studies, participants receiving group education by health care professionals had a 33% reduced diabetes risk (odds ratio 0.67 [0.49, 0.92]). Intervention participants lost 1.5 kg more weight and achieved a 0.09 mmol/L greater FPG decrease than control participants. Every additional kilogram lost by participants was associated with 43% lower diabetes odds (b = 0.57 [0.41,
0.78]). In the proof-of-concept trials the absolute risk reduction has been approximately 17%. Reasons for the lower absolute benefits in real-world studies compared with the proof-of-concept trials are several: differences in entry criteria, intensity and design of interventions, duration of interventions, etc.
Galaviz KI, Weber MB, Straus A, Haw JS, Narayan KMV, Ali MK. Global Diabetes Prevention Interventions: A Systematic Review and Network Meta-analysis of the Real-World Impact on Incidence, Weight, and Glucose. Diabetes Care. 2018;41:1526-1534.
- The UK national program should receive greater attention, as it has been developed within primary care in a national health system, has achieved a relatively large reach among those at high risk, and has published preliminary outcome evaluations.
Authors reply:
We have added text on the UK national program as follows:
“A recent review included 65 articles reporting on the English NHS DPP since 2015 were eligible for inclusion, most of them published during 2018 to 2020 (Whelan and Bell 2022). The articles reported on uptake and retention, implementation considerations, program outcomes, stakeholder experience and screening and referral processes, various research methods. Articles revealed preliminary evidence on service user characteristics, rates of referral, uptake and retention as well as how far the NHS DPP is being delivered in line with its evidence base and service specification. The evidence is gradually accumulating on NHS DPP uptake and retention most, with emerging evidence on program outcomes (such as weight loss and HbA1c).”
Whelan M, Bell L. The English national health service diabetes prevention programme (NHS DPP): A scoping review of existing evidence. Diabet Med. 2022;39:e14855.
- Line 112/292: In terms of the question of lesser effectiveness in those identified at high risk by IFG rather than IGT, please add references.
Authors reply:
These references are useful and have been added to the text.
“The Diabetes Community Lifestyle Improvement Program (CLIP) there was evidence for heterogeneity of the intervention effect across prediabetes type, with the strongest benefit in people with combined IFG+IGT (36%) and isolated IGT (31%) and not significant in isolated IFG (12%). (Weber et al.) The randomized controlled trial, the Kerala Diabetes Prevention Program, a low-cost community-based peer-support lifestyle intervention where participants were identified on the basis of a risk score, and the majority had isolated IFG or NGT resulted in a nonsignificant reduction in diabetes incidence (Thankappan et al.).”
Weber MB, Ranjani H, Staimez LR, et al. The stepwise approach to diabetes prevention: results from the D-CLIP randomized controlled trial. Diabetes Care. 2016;39:1760-1767.
Thankappan KR, Sathish T, Tapp RJ, et al.: A peer-support lifestyle intervention for preventing type 2 diabetes in India: A cluster-randomized controlled trial of the Kerala Diabetes Prevention Program. PLoS Med. 2018;15(6): e1002575.
- Line 130: It seems unbalanced to mention the long-term benefit seen in the Da Qing trial but not to provide the discouraging numbers from the DPP in terms of cardiovascular and mortality outcomes.
Authors reply:
Both the DPS and DPP have assessed CVD and mortality outcomes. We have added the following text:
“ In DPS, total mortality and CVD incidence did not differ significantly between the intervention and control groups during the extended 10-year follow-up (Uusitupa 2009). However, compared with the population-based cohort with IGT, adjusted HRs were 0.21 (95% CI 0.09–0.52) and 0.39 (0.20–0.79) for total mortality, and 0.89 (0.62–1.27) and 0.87 (0.60–1.27) for cardiovascular morbidity in the intervention and control groups, respectively. The risk of death in DPS combined cohort was markedly lower than in the Finnish IGT cohort (adjusted HR 0.30, 95% CI 0.17–0.54), while the difference in the risk of CVD was smaller (adjusted HR 0.88, 95% CI 0.64–1.21). A large proportion of the DPS participants received evidence-based management for hypertension and dyslipidemia both in the intervention and control group, since they were advised to contact their physician if values were too high at the annual visit. In DPPOS, neither metformin nor lifestyle intervention reduced major cardiovascular events over 21 years despite long-term prevention of diabetes (Goldberg et al.). Provision of group lifestyle intervention to all, extensive out-of-study use of statin and antihypertensive agents, and reduction in the use of study metformin together with out-of-study metformin use over time have diluted the effects of the DPP interventions. Also, since the modern management of T2D has emphasized a multifactorial treatment of people with diabetes, in the control groups of the DPS and DPP have received treatment for hypertension and dyslipidemia once they had been diagnosed with T2D, thus earlier than in the intervention group where people were free of T2D longer time.”
Uusitupa M, Peltonen M, Lindström J, Aunola S, Ilanne-Parikka P, Keinänen-Kiukaanniemi S, Valle TT, Eriksson JG, Tuomilehto J; Finnish Diabetes Prevention Study Group. Ten-year mortality and cardiovascular morbidity in the Finnish Diabetes Prevention Study--secondary analysis of the randomized trial. PLoS One. 2009;4:e5656.
Goldberg RB, Orchard TJ, Crandall JP, Boyko EJ, Budoff M, Dabelea D, Gadde KM, Knowler WC, Lee CG, Nathan DM, Watson K, Temprosa M; Diabetes Prevention Program Research Group*. Effects of Long-term Metformin and Lifestyle Interventions on Cardiovascular Events in the Diabetes Prevention Program and Its Outcome Study. Circulation. 2022May 31;145(22):1632-1641
- Line 440: At some point, perhaps in this section, the lack of population reach of high-risk strategies and thus their relevance to the overall reduction of risk of diabetes in populations should be addressed.
Authors reply:
This is a valid point and we have added the following text:
Several real-world studies on T2D prevention have demonstrated an important challenge with the high-risk approach in NCD prevention: the identification of the high-risk status does not lead to the adoption of services available for the risk reduction. For instance, in the English NHS DPP 324,699 people were referred into the program of whom 152,294 attended an initial assessment and only 96,442 (30%) attended at least one of the group-based intervention sessions (Valabhji). In the Finnish D2D study, 10,149 individuals at high risk for T2D were identified primary health care centers, but one-year follow-up data were available only for 2,798 (28%) participants (Saaristo).
Saaristo T, Moilanen L, Korpi-Hyövälti E, et al. Lifestyle intervention for prevention of type 2 diabetes in primary health care: one-year follow-up of the Finnish national diabetes prevention program (FIN-D2D). Diabetes Care 2010; 33: 2146-51.
- Line 456: Why should SSB taxes not be useful in LMICs?
Authors reply:
We agree with the reviewer on this comment and have deleted this sentence and added the following text:
“ It has been proposed that taxation may be a cost-effective strategy for reducing sugar intake, obesity, and health costs and generating revenue that can be used for health programs (Brownell). An excise tax of one peso per liter on sugar-sweetened beverages was implemented in Mexico in 2014 (Basto). A cohort simulation model was used to project the impact of the tax over ten years. The tax was estimated to prevent 239,900 cases of obesity and 61,340 cases of T2D, leading to gains of 55,300 quality-adjusted life-years, and averting 5,840 disability-adjusted life-years. The tax was estimated to save USD 3.98 per one dollar spent on its implementation.
Brownell KD, Farley T, Willett WC, et al. The public health and economic benefits of taxing sugar-sweetened beverages. N Engl J Med. 2009;361:1599–605
Basto-Abreu A, Barrientos-Gutiérrez T, Vidaña-Pérez D, et al. Cost-Effectiveness Of The Sugar-Sweetened Beverage Excise Tax In Mexico. Health Aff. 2019;38:1824-1831.
- Line 490: The title of this section begins "Who are the main stakeholders...". However, few are listed. Stakeholders would logically also include the international food and beverage companies against whose products taxes and advertising restrictions are being suggested and local farmers whose products might receive subsidies to stimulate their consumption. As these companies won't accept changes without a struggle and given lobbying with limited restrictions is permitted around the world, not just government officials who work in health ministries and departments but also politicians and NGOs become relevant stakeholders.
Authors reply:
We have added text to this issue as follows:
“Stakeholders would also include the local and multinational food and beverage companies against whose products taxes and advertising restrictions are being suggested. Farmers who may produce health promoting foods might receive subsidies to grow such foods, and their consumption should be also promoted by retailers. Politicians and non-governmental organizations are also relevant stakeholders. An ecological analysis in California found that commuters with the highest distance and time traveled also had the highest rates of physical inactivity and obesity (López-Zetina et al., 2006). Similarly, longer travel distance and time have been associated with higher BMI and waist circumference (Frank et al., 2004; Hoehner et al., 2012). The Finnish prospective study showed that commuting physical activity was inversely associated with the incidence of T2D (Hu).It is commonly agreed that modern city planning should take into account health issues to make physical activities convenient, easy and safe.”
Lopez-Zetina J, Lee H, Friis R. The link between obesity and the built environment. Evidence from an ecological analysis of obesity and vehicle miles of travel in California. Health Place. 2006;12:656–664
Hoehner CM, Barlow CE, Allen P, Schootman M. Commuting distance, cardiorespiratory fitness, and metabolic risk. Am J Prev Med. 2012;42:571–578.
Hu G, Qiao Q, Silventoinen K, et al. Occupational, commuting, and leisure-time physical activity in relation to risk for Type 2 diabetes in middle-aged Finnish men and women. Diabetologia 2003; 46: 322-9.
- Line 523: A global treaty sounds like an interesting idea. However, to suggest it without detailing what would be involved seems questionable. Would there be limitations parallel to those in the tobacco convention framework on advertising and taxes on the consumption of unhealthy foods and beverages? Prohibition of sales of fast foods in or near schools? I suggest furnishing details of what you think should go into such a treaty or deleting this section.
Authors reply:
We find this suggestion very important since it will be important to have countries more committed to tackle the diabetes epidemic, and hope to keep this section. Obviously, at this point it is not possible for us the design such a treaty. Nevertheless, we have expanded this section with some suggestions as follows:
“To develop such a treaty, a convention needs to be organized. The implementation its
provisions shall be guided by clearly defined principles that may include as done for the WHO FCTC:
- Information of the health consequences of T2D and measures at the appropriate governmental level to protect all persons from exposure to risk factors for T2D;
- Strong political commitment is necessary to develop and support, at the national, regional and international levels, and comprehensive multisectoral measures;
- International cooperation, particularly transfer of technology, knowledge and financial
assistance and provision of related expertise, taking into consideration local culture, as well as social, economic, political and legal factors;
- Comprehensive multisectoral measures in accordance with evidence-based public health principles;
- Technical and financial assistance addressed in the context of nationally developed strategies for sustainable development.
- The participation of civil society is essential in achieving the objective of the treaty.”
Minor Comments
- Much of what you cover in the manuscript goes beyond National Intervention Programs. Might the title be better as: "Type 2 ....to National intervention and Beyond"?
We agree and changed the title accordingly.
- Introduction, Line 44: The term evidence-based medicine arose around 1990. Thus, is it not more appropriate to say"...only develop momentum in the 1990s"?
We agree and changed the text accordingly.
- Line 200. Figure 1 is missing—only its legend appears in the draft I received.
We are sorry and have now submitted the Figure.
- Line 237: Add a reference supporting the size of weight loss in the US DPP Program (e.g., http://dx.doi.org/10.5888/pcd16.190053)
We agree and have added this reference.
Chakkalakal RJ, Connor LR, Rolando LA, et al. Putting the National Diabetes Prevention Program to Work: Predictors of Achieving Weight-Loss Goals in an Employee Population. Prev Chronic Dis. 2019;16:E125.
- Line 363: Something wrong with this sentence.
Corrected “Due to the fact that T2D and prediabetes are asymptomatic conditions people with such disorders may consider interventions unnecessary.
- Line 425: The discussion of dietary fiber seems out of place in this section on precision medicine.
We find the issues of dietary fiber important for precision medicine on T2D prevention and hope to keep this section. We have stated that in most populations fiber intake is too low and since low fiber intake is associated with T2D, therefore, it is an essential part of the prevention of T2D.
Reviewer 2 Report (Previous Reviewer 2)
The authors have properly clarified all issues.
Author Response
Reviewer 2.
- In the section about RCT in T2DM prevention, it is important to clarify that the definition of impaired glucose tolerance itself has changed along the years, making difficult to compare these trials among themselves. I also suggest to the authors state what is the current definition for IGT at this section, so the reader may become more familiar with this concept.
Authors reply:
We think that the reviewer is confused with the definition of impaired fasting glucose (IFG), since the definition of IGT has remained unchanged since its launch by the WHO Expert Group in 1980: 2-h post-challenge glucose 7.8-11.0 mmol/L.
2- The DPP trial is the largest one in T2DM prevention, so that its estimates of effect are expected to be more precise compared to other trials. I suggest to reinforce that in this section.
Authors reply:
It is correct that the DPP with its largest trial population had more precise estimates of the overall effect of the intervention. However, the results with lifestyle intervention were identical to those in other trials in people with IGT. Actually, the DPP results were confirmatory to those of the Finnish DPS, and therefore should not be overemphasized. The confidence intervals in effect estimates do not differ between the DPS and DPP.
3- The sentence “This estimate was strongly influenced by the 6% only 120 risk reduction during the DPP Outcomes Study (DPPOS) where all participants regardless of their original treatment group were offered lifestyle counseling (21)” seems inaccurate. It is expected that, after a successful RCT, patients randomized to the control arm should be offered the possibility of receiving the intervention, given it is established as effective.
Authors reply:
We believe that our original statement was correct. It is also true that once a RCT finds positive outcome with a certain intervention, it should be offered also to those who were originally allocated the control group. Such management is usually left to the usual health care and not done by the trial personnel who are not part of the local health providers. This was the way how in the Finnish DPS the follow-up was carried out. However, in the US DPP, the actual trial personnel provided additional management after the trial was over which is unusual and obviously confounded the follow-up outcome.
4- I suggest to replace the term “real life” settings at lines 68, 158,256 and 389 for “outside of randomized studies” setting. The term “real-life” or “real-world” seems misleading since it may imply that RCTs are conducted with “unreal” patients.
Authors reply:
This is a semantic question. A lot of papers published have been using the term “real-life”. We need to ask what the style of this journal is. “Outside of randomized studies” sounds complex and is not at all clear either. We prefer to keep our original wording.
5- The FINDRISC score is really an important initiative and worthy mentioning. Could the authors comment briefly on studies that have externally validated this score outside Finland or European countries?
Authors reply:
As we have stated over 200 publications in the PubMed can be found where the FINDRISC is mentioned in the title of the publication. We agree that the externally validation of this score outside Finland or European countries is a useful issue and should be taken up in the future. However, it is another topic for a new report in the future.
6- In the US National Diabetes Prevention Program, do the authors have the percentage of black or Latino patients who were covered? This issue is of great importance to assess how the program was inclusive.
Authors reply:
No, we do not have this information. This may not be possible due to legal/ethical regulations.
7- Can the authors mention about impact (if any) of the diabetes prevention programs on CV and all-cause mortality?
Authors reply:
We have added more information on this issue. The section reads now:
“Questions remain about the degree to which lifestyle interventions for diabetes prevention also convert into reduced risk of diabetes complications. None of the proof-of-concept trials were statistically powered to study cardiovascular (CVD) outcomes. Thus, any post-hoc findings must be reviewed with caution. The Da Qing diabetes prevention follow-up study found reduced rates of CVD events and deaths and composite microvascular events 24 years after the completion of the intervention among persons in the interventions (22). In the DPS the occurrence of early retinopathy changes, i.e., microaneurysms were significantly lower in the intervention than in the control group (29). In DPS, total mortality and CVD incidence did not differ significantly between the intervention and control groups during the extended 10-year follow-up (30). However, compared with the population-based cohort with IGT, adjusted HRs were 0.21 (95% CI 0.09–0.52) and 0.39 (0.20–0.79) for total mortality, and 0.89 (0.62–1.27) and 0.87 (0.60–1.27) for cardiovascular morbidity in the intervention and control groups, respectively. The risk of death in DPS combined cohort was markedly lower than in the Finnish IGT cohort (adjusted HR 0.30, 95% CI 0.17–0.54), while the difference in the risk of CVD was smaller (adjusted HR 0.88, 95% CI 0.64–1.21). A large proportion of the DPS participants received evidence-based management for hypertension and dyslipidemia both in the intervention and control group, since they were advised to contact their physician if values were too high at the annual visit. In DPPOS, neither metformin nor lifestyle intervention reduced major cardiovascular events over 21 years despite long-term prevention of diabetes (31). Provision of group lifestyle intervention to all, extensive out-of-study use of statin and antihypertensive agents, and reduction in the use of study metformin together with out-of-study metformin use over time have diluted the effects of the DPP interventions. Also, since the modern management of T2D has emphasized a multifactorial treatment of people with diabetes, in the control groups of the DPS and DPP have received treatment for hypertension and dyslipidemia once they had been diagnosed with T2D, thus earlier than in the intervention group where people were free of T2D longer time.
While the issue of preventive effect on complications remains under debate, future T2D prevention studies should also consider the possibility that some of the benefits of lifestyle intervention will lie in more diverse outcomes, such as disability, sleep apnea, quality of life, depression, and mental health (32).
8- I would add at the final of the section about low to middle income countries that an additional challenge is the high heterogeneity of health access within those countries. In Brazil and Mexico, for example, there may exist places with per capita income similar to US and Europe, whereas other cities/villages with social-economic status resembling the poorest countries in the world.
Authors reply:
We agree that this is an important issue, but it is very complex and cannot be handled with a few words or sentences, and how it is related to the national diabetes prevention programs is not clear.
9- In the section “T2D prevention in youth”, please, define what the authors mean as “youth”. Younger than 40 years? Or do you mean teenagers/adolescents?
Authors reply:
Youth is a general term, and its interpretation is flexible. We do not find a reason to start writing about the definition of such a term. Our references give a good evidence what we have been meaning with this.
10- The last paragraph from the section “5.6. Precision medicine in the prevention of T2D” fits better in the section about diet and lifestyle strategies. The sentence “..in general glucose-lowering drugs should not be apriority in national interventions to prevent T2D” (lines 473-473) seems inaccurate since metformin has been demonstrated to be effective in preventing T2DM, although with modest benefit. I recommend to remove or modify.
Authors reply:
Precision medicine is its own topic and not only diet and lifestyle. We would like to keep our original statement about glucose-lowering drugs including metformin. Although they do lower blood glucose (also in people with prediabetes) by definition, it has been demonstrated that if the use of such drugs in people at high risk of T2D is stopped, the incidence of diabetes is getting to the level of that without such treatment.
Minor issues:
- At lines 82-83, it reads “The Da Qing Diabetes Prevention Study carried out in northern China the 33 participating clinics were randomized…” This sentence may have some spelling issue. Do the authors mean “The Da Qing Diabetes Prevention Study carried out in northern China had 33 participating clinics that were randomized..”?
Authors reply:
Thanks for this clarification, we have changed the sentence accordingly.
Reviewer 3 Report (Previous Reviewer 3)
No comments.
Author Response
Minor revision
This study discussed the need for a national intervention to prevent T2D. This would be beneficial for people suffering from prediabetes, impaired glucose tolerance, unhealthful obesity, etc.
However, from my perspective, I have some following comments and suggestions:
- What is "he" on line 82?”. Please re-check.
Authors’ reply. Thank you for finding this typo. This sentence has been changed to: “The Da Qing Diabetes Prevention Study carried out in northern China had 33 participating clinics that were randomized…”
- Line 165: typo error
Authors’ reply. This full stop has been changed to comma “….in the entire country, an….”
- The full term NCD should be defined in the text. If the abbreviations appear in the first place, the authors should ensure that they are defined.
Authors’ reply. This has been done “non-communicable disease (NCD)”
- The conclusion paragraph should be included in this review, such as " Conclusion" so that readers can catch the primary findings and suggestions after reading the manuscript.
Authors’ reply. We think that this is sufficiently done in the Abstract and does not need to be repeated at the end of the paper. If the Editor finds such a conclusion paragraph necessary, we can naturally insert it, but as said it would have almost the same text as given in the Abstract.
- It seems that the authors are responsible for defining what a typical review article might be called, like "narrative reviews", from what I could tell.
Authors’ reply. This is correct, the paper is a “narrative review”.
This manuscript is a resubmission of an earlier submission. The following is a list of the peer review reports and author responses from that submission.
Round 1
Reviewer 1 Report
I am honored to have the opportunity to review your manuscript – a work written by leaders in the field of diabetes prevention. I hope that the comments below can help strengthen the final product.
Major Comments
1. Line 208: High-risk strategy interventions, when translated to the community or primary care settings, have demonstrated decreased effectiveness (25-30% in meta-analyses) when compared to the almost 60% risk reduction seen in the original trials. The probability of those entering most community or national programs to develop diabetes, given different entry criteria from the initial IGT trials, is generally lower than in the original trials. These two factors will result in a considerably lower absolute benefit for those entering these programs. The Galaviz review (Galaviz KI, Weber MB, Straus A, Haw JS, Narayan KMV, Ali MK. Global Diabetes Prevention Interventions: A Systematic Review and Network Meta-analysis of the Real-World Impact on Incidence, Weight, and Glucose. Diabetes Care. 2018 Jul;41(7):1526-1534.) suggested an absolute benefit in reduction of diabetes over the first 1-2 years of just 3%. This difference from the original studies makes for less attractive cost-effectiveness analyses than reported using effectiveness data from the initial trials. Though short follow-up might explain part of the low effectiveness reported, the average DPP follow-up, in comparison, was 2.8 years. This issue must be addressed.
2. The UK national program should receive greater attention, as it has been developed within primary care in a national health system, has achieved a relatively large reach among those at high risk, and has published preliminary outcome evaluations.
3. Line 112/292: In terms of the question of lesser effectiveness in those identified at high risk by IFG rather than IGT, please add references to the Diabetes Community Lifestyle Improvement Program (Weber MB, Ranjani H, Staimez LR, et al. The stepwise approach to diabetes prevention: results from the D-CLIP randomized controlled trial. Diabetes Care. 2016;39:1760-1767) and the Kerala Diabetes Prevention Program (Thankappan KR, Sathish T, Tapp RJ, et al.: A peer-support lifestyle intervention for preventing type 2 diabetes in India: A cluster-randomized controlled trial of the Kerala Diabetes Prevention Program. PLoS Med. 2018; 15(6): e1002575).
4. Line 130: It seems unbalanced to mention the long-term benefit seen in the DaQing trial but not to provide the discouraging numbers from the DPP in terms of cardiovascular and mortality outcomes (Goldberg RB, Orchard TJ, Crandall JP, Boyko EJ, Budoff M, Dabelea D, Gadde KM, Knowler WC, Lee CG, Nathan DM, Watson K, Temprosa M; Diabetes Prevention Program Research Group*. Effects of Long-term Metformin and Lifestyle Interventions on Cardiovascular Events in the Diabetes Prevention Program and Its Outcome Study. Circulation. 2022 May 31;145(22):1632-1641.). The results from the DPP do not appear to reflect just a lack of power.
5. Line 440: At some point, perhaps in this section, the lack of population reach of high-risk strategies (See: Roberts S, Barry E, Craig D, Airoldi M, Bevan G, Greenhalgh T. Preventing type 2 diabetes: systematic review of studies of cost-effectiveness of lifestyle programmes and metformin, with and without screening, for pre-diabetes. BMJ Open. 2017 Nov 15;7(11):e017184 and Venkataramani M, Pollack CE, Yeh HC, Maruthur NM. Prevalence and Correlates of Diabetes Prevention Program Referral and Participation. Am J Prev Med. 2019 Mar;56(3):452-457.) and thus their relevance to the overall reduction of risk of diabetes in populations should be addressed.
6. Line 456: Why should SSB taxes not be useful in LMICs? SSBs have conquered consumers worldwide and are far from just a high-income country phenomenon. Significant reductions in SSB consumption resulting from these taxes have been documented in Mexico.
7. Line 490: The title of this section begins "Who are the main stakeholders...". However, few are listed. Stakeholders would logically also include the international food and beverage companies against whose products taxes and advertising restrictions are being suggested and local farmers whose products might receive subsidies to stimulate their consumption. As these companies won't accept changes without a struggle and given lobbying with limited restrictions is permitted around the world, not just government officials who work in health ministries and departments but also politicians and NGOs become relevant stakeholders.
8. Line 523: A global treaty sounds like an interesting idea. However, to suggest it without detailing what would be involved seems questionable. Would there be limitations parallel to those in the tobacco convention framework on advertising and taxes on the consumption of unhealthy foods and beverages? Prohibition of sales of fast foods in or near schools? I suggest furnishing details of what you think should go into such a treaty or deleting this section.
Minor Comments
1. Much of what you cover in the manuscript goes beyond National Intervention Programs. Might the title be better as: "Type 2 ....to National intervention and Beyond"?
2. Introduction, Line 44: The term evidence-based medicine arose around 1990. Thus, is it not more appropriate to say "...only develop momentum in the 1990s"?
3. Line 200. Figure 1 is missing—only its legend apprears in the draft I received.
4. Line 237: Add a reference supporting the size of weight loss in the US DPP Program (e.g., http://dx.doi.org/10.5888/pcd16.190053external icon)
5. Line 363: Something wrong with this sentence.
6. Line 425: The discussion of dietary fiber seems out of place in this section on precision medicine.
Reviewer 2 Report
This review paper from Tuomilehto et al. brings important summary of the current knowledge about interventions to prevent T2DM. The paper is well written (although there are minor typos) and the topic is timely and very relevant, as T2DM is estimated to be the next “pandemic” to ravage lives all over the world. Some suggestions/remarks follow:
1- In the section about RCT in T2DM prevention, it is important to clarify that the definition of impaired glucose tolerance itself has changed along the years, making difficult to compare these trials among themselves. I also suggest to the authors state what is the current definition for IGT at this section, so the reader may become more familiar with this concept.
2- The DPP trial is the largest one in T2DM prevention, so that its estimates of effect are expected to be more precise compared to other trials. I suggest to reinforce that in this section.
3- The sentence “This estimate was strongly influenced by the 6% only 120 risk reduction during the DPP Outcomes Study (DPPOS) where all participants regardless of their original treatment group were offered lifestyle counseling (21)” seems inaccurate. It is expected that, after a successful RCT, patients randomized to the control arm should be offered the possibility of receiving the intervention, given it is established as effective.
4- I suggest to replace the term “real life” settings at lines 68, 158, 256 and 389 for “outside of randomized studies” setting. The term “real-life” or “real-world” seems misleading since it may imply that RCTs are conducted with “unreal” patients.
5- The FINDRISC score is really an important initiative and worthy mentioning. Could the authors comment briefly on studies that have externally validated this score outside Finland or European countries?
6- In the US National Diabetes Prevention Program, do the authors have the percentage of black or Latino patients who were covered? This issue is of great importance to assess how the program was inclusive.
7- Can the authors mention about impact (if any) of the diabetes prevention programs on CV and all-cause mortality?
8- I would add at the final of the section about low to middle income countries that an additional challenge is the high heterogeneity of health access within those countries. In Brazil and Mexico, for example, there may exist places with per capita income similar to US and Europe, whereas other cities/villages with social-economic status resembling the poorest countries in the world.
9- In the section “T2D prevention in youth”, please, define what the authors mean as “youth”. Younger than 40 years? Or do you mean teenagers/adolescents?
1- The last paragraph from the section “5.6. Precision medicine in the prevention of T2D” fits better in the section about diet and life style strategies.
1- The sentence “..in general glucose-lowering drugs should not be a priority in national interventions to prevent T2D” (lines 473-473) seems inaccurate since metformin has been demonstrated to be effective in preventing T2DM, although with modest benefit. I recommend to remove or modify.
Minor issues:
1- At lines 82-83, it reads “The Da Qing Diabetes Prevention Study carried out in northern China he 33 participating clinics were randomized…” This sentence may have some spelling issue. Do the authors mean “The Da Qing Diabetes Prevention Study carried out in northern China had 33 participating clinics that were randomized..”?
Reviewer 3 Report
Minor revision
This study discussed the need for a national intervention to prevent T2D. This would be beneficial for people suffering from prediabetes, impaired glucose tolerance, unhealthful obesity, etc.
However, from my perspective, I have some following comments and suggestions:
1. What is "he" on line 82?”. Please re-check.
2. Line 165: typo error
3. The full term NCD should be defined in the text. If the abbreviations appear in the first place, the authors should ensure that they are defined.
4. The conclusion paragraph should be included in this review, such as "6. Conclusion" so that readers can catch the primary findings and suggestions after reading the manuscript.
5. It seems that the authors are responsible for defining what a typical review article might be called, like "narrative reviews", from what I could tell.